# A Novel GB-SAR System Based on TD-MIMO for High-Precision Bridge Vibration Monitoring

Zexi Zhang [1], Zhiyong Suo [1,*], Feng Tian [2], Lin Qi [1], Haihong Tao [1] and Zhenfang Li [1]

1 National Key Laboratory of Radar Signal Processing, Xidian University, Xi'an 710071, China
2 Nanjing Research Institute of Electronics Technology, Nanjing 210039, China
* Correspondence: zysuo@xidian.edu.cn; Tel.: +86-137-2062-2996

**Abstract:** Ground-based synthetic aperture radar (GB-SAR) is a highly effective technique that is widely used in landslide and bridge deformation monitoring. GB-SAR based on multiple input multiple output (MIMO) technology can achieve high accuracy and real-time detection performance. In this paper, a novel method is proposed to design transmitting and receiving array elements, which increases the minimum spacing of the antenna by sacrificing several equivalent phase centers. In MIMO arrays, the minimum antenna spacing in the azimuth direction is doubled, which increases the variety of antenna options for this design. To improve the accuracy of the system, a new method is proposed to estimate channel phase errors, amplitude errors, and position errors. The position error is decomposed into three directions with one compensated by the phase error and two estimated by the strong point. Finally, we validate the accuracy of the system and our error estimation method through simulations and experiments. The results prove that the GB-SAR system performs well in bridge deformation and vibration monitoring with the proposed method.

**Keywords:** MIMO imaging radar; GB-SAR array arrangement; array error estimation; bridge vibration; deformation detection





## 1. Introduction

In recent decades, the research on ground-based synthetic aperture radars (GB-SAR) in bridge detection has blossomed. The high accuracy measurement capability of GB-SAR makes it an effective technology for landslide and deformation monitoring [1–6]. For example, typical radar systems LISA [7,8] and Fast-SAR [9–11] have been proven effective in these fields. Generally, data acquisition time and imaging time are important parameters for ground-based detection systems [12]. Conventional mechanical scanning GB-SARs with single or two transceiver elements rely on external motion to obtain a large aperture. Although the GB-SAR scanning time has been reduced to several minutes to acquire one set of data, it cannot handle the time-critical application cases, such as bridge vibration [13] and building swing measurement.

In recent years, multiple-input multiple-output (MIMO) technology [14] has gradually been applied to GB-SAR systems [15–17]. MIMO radars can achieve larger synthetic aperture using fewer transceiver array elements [18–20]. Generally, MIMO system implementation methods are divided into three types, including waveform diversity (WD), frequency diversity (FD) and time division multiplexing (TDM) [21]. The first method is WD, where orthogonal waveforms are emitted by transmitters and recovered at the reception. Nevertheless, large bandwidth and fully orthogonal emission waveforms are hard to produce. The second method is FD, where signals of different frequencies are radiated from the transmitter while being completely separated from the receiver. This method requires a high manufacturing cost for simultaneous receiving, although the data acquisition time is short. The third one is the TDM method, where the transmitting and receiving elements are enabled by a switch matrix [22,23]. The implementation of this method is easier than

WD and FD because it avoids the problem of waveform and frequency separation. The majority of MIMO radar systems operate in TDM mode, for example, the MIMO-based non-collinear array system [24], MELISSA system [25], 77G Millimeter wave system [26], and the Ku-Band MIMO GB-SAR system [19]. TDM-MIMO (referred to as MIMO except for special instructions) GB-SAR can compensate for the disadvantages of low data acquisition rate and motion error [27] in mechanical scanning GB-SAR.

The spacing between the transmitting elements or receiving elements is a critical parameter in the design of MIMO transceiver arrays. The grating lobe will appear if the spacing is too large. Otherwise, the available types of antennas will be limited by the small spacing. For example, the MELISSA system [25] utilizes two different types of antennas (Horn and Vivaldi) to design transceiver elements. Vivaldi antennas [28] with small physical dimensions in the horizontal direction are selected to accommodate the narrow element spacing. However, the weak structure of Vivaldi antennas is not appropriate for the outdoor environment. The low gain and maximum power tolerance are the obvious deficiencies of Vivaldi antennas compared with Horn antennas. To solve this problem, a new array of elements arrangement method is proposed. This method can increase the minimum spacing for antenna placement. Horn antennas with the same beamwidth are designed for transmitting and receiving. In addition, the Horn antenna can achieve high radiation gain, which improves the detection distance with the same transmitting power.

Furthermore, during production and installation, channel amplitude errors, phase errors, and position errors appear because of the specific multi-antenna structure of the MIMO radar. There are two types of phase errors, the first one is the phase difference caused by the inconsistent characteristics of the radar RF components [29], and the second one is the phase difference caused by the radar installation position [30]. Array error estimation methods have been comprehensively researched. The method proposed in [31] estimates the channel-to-channel amplitude and phase errors based on a single target. The error between the azimuth signal processing results and the ideal results is minimized by adjusting channel weights. However, this method can only calibrate the inter-channel error, which ignores the intra-channel error calibration. Reference [29] proposes a near-field calibration method using an adaptive weighting technique. The method optimizes the channel weights to make the peak of the main lobe ideal. This method ignores the phase errors caused by the position shift of the array elements. Reference [32] proposes a multi-angle anti-calibration method suitable for MIMO radars. This method is not compatible with high-speed GB-SAR because of the long estimation and calibration time. To solve these issues above, a high-accuracy error estimation method based on this system is proposed. In this method, the position error of the array elements is decomposed into three different directions according to the coordinates. The distance direction position error is regarded as a part of the initial phase errors. Horizontal direction and vertical direction position errors are estimated with multiple strong reflect targets.

In addition, a large number of experimental implementations are carried out to verify the system. The experiments are divided into three parts. In the first part, the correctness of the transceiver array element arrangement is verified. Furthermore, the validity of the error estimation method is verified by image entropy. The second part is a pre-set programmable moving-corner reflector experiment. The high accuracy deformation detection capability and fast deformation detection capability of the system are confirmed in this experiment. The third part of the experiment is an actual bridge measurement case. When vehicles pass the bridge, deformation with the frequency information of the bridge can be observed by the system. The advantage of this GB-SAR is that the deformation information of all points in the scene can be detected simultaneously.

The remaining contents of this paper are organized as follows. Section 2 shows the architecture of the GB-SAR system. Section 3 introduces the basic principles of EPC and establishes the radar signal model. Section 4 discusses the arrangement design of the new MIMO array. Section 5 describes the estimation and calibration of the channel amplitude errors, phase errors, and array element position errors. Section 6 presents the relationships

between imaging results and system errors, such as channel amplitude errors, phase errors, and position errors. Image entropy is used to evaluate the experimental results before and after array calibration. Section 7 verifies the high-frequency vibration monitoring capability of the new ground-based synthetic aperture radar. The experimental results of vibration and deformation monitoring of bridges are analyzed. Sections 8 and 9 are the discussion and conclusion of the article.

## 2. The TD-MIMO GB-SAR System Architecture

GB-SAR can achieve high-accuracy and real-time detection performance. The system consists of array arrangement calculation and design, system error estimation, imaging process, selecting the interest area, and deformation analysis, as shown in Figure 1. In the array arrangement calculation and design, we redesign transmitting and receiving array elements, which increases the minimum spacing of the antenna by sacrificing several equivalent phase centers. In system error estimation, we propose a new method to estimate channel phase, amplitude errors, and position errors. The error compensation can improve the accuracy of the system. We use the back projection (BP) algorithm for the imaging process. To reduce the calculation rate of data processing, only the interest area is selected for observation. At last, the corresponding deformation curve is analyzed.

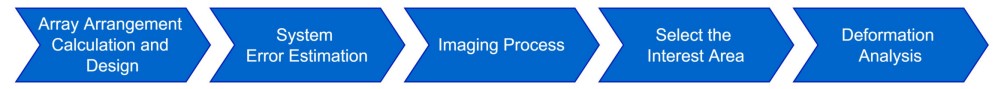

**Figure 1.** The TD-MIMO GB-SAR system design and bridge vibration monitoring procedure.

The TD-MIMO GB-SAR system architecture procedure is as follows. Firstly, compared with the traditional array, we increase the minimum spacing for antenna placement, which makes it possible to choose Horn antennas for transmitting and receiving. Next, we propose a new error estimation algorithm applied to the newly designed array. Error estimation results play an important role in BP imaging, which can improve the quality of the image. The interest area or interest point can be selected for observation. The deformations of the interest area are expressed in the form of phase change. Finally, the time and frequency domain information can be obtained by deformation analysis.

## 3. Array Element Equivalence Principle and Signal Model

### 3.1. Array Element Equivalence Principle

In the Cartesian coordinate system, it is assumed that the position of the $n$th transmitting unit is $T_n(x, y, z), n = 1, 2, \ldots, N$. The position of the $m$th receiving unit is $R_m(x, y, z), m = 1, 2, \ldots, M$. $A(x, y, z)$ is the coordinate of the target. The propagation path of the electromagnetic (EM) wave is from the $n$th transmitting unit to the target and returns to the $m$th receiving unit. The corresponding distance is as follows:

$$r_{T_n, R_m}^A = r_{T_n}^A + r_{R_m}^A, \tag{1}$$

where

$$r_{T_n}^A = \sqrt{(x_{T_n} - x_A)^2 + (y_{T_n} - y_A)^2 + (z_{T_n} - z_A)^2}, \tag{2}$$

$$r_{R_m}^A = \sqrt{(x_{R_m} - x_A)^2 + (y_{R_m} - y_A)^2 + (z_{R_m} - z_A)^2}. \tag{3}$$

$r_{T_n}^A$ represents the distance from the target to the $n$th transmitting array element. $r_{R_m}^A$ represents the distance from the target to the $m$th receiving array element.

If the distance from the target to the radar is much larger than the distance between the transmitting array elements and the receiving array elements [33], it can be approximated as follows:

$$r_{T_n}^A + r_{R_m}^A \approx 2r_{P_{n,m}}^A, \tag{4}$$

where $P_{n,m} = \frac{1}{2}(T_n + R_m)$ is the barycenter of two phase centers. In the far-field condition, the transmitting array elements and the receiving array elements are equivalent to a virtual transceiver center, which is called the equivalent phase center (EPC).

　　The arc line in Figure 2 is the spherical wave passing through the EPCs. In near-field conditions, the echo phase between the receiving array element and the EPCs point is not consistent. The curvature of the wavefront arc decreases by increasing the target distance $R$ under the condition that the distance between the transmitting and receiving array elements remains constant. When the phase difference between the transceiver elements and the EPCs is less than $\pi/4$, it is regarded as the boundary condition between the far and near fields. The boundary condition can be expressed as:

$$r_{T_n}^A + r_{R_m}^A - 2r_{P_{n,m}}^A \leq \frac{\lambda}{4}, \tag{5}$$

where $\lambda$ is the carrier wavelength.

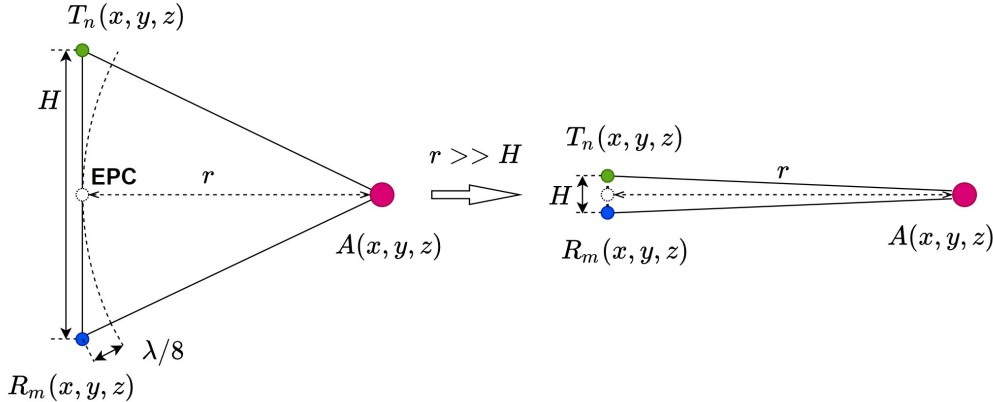

**Figure 2.** The illustration of equivalent transmitting and receiving array elements.

*3.2. Signal Model*

　　Assume that the FMCW signal emitted by the radar is described as:

$$s_{T_n}(t) = \mathrm{rect}\left(\frac{t}{T_P}\right)\exp\left\{j\pi\left(2f_c t + K_r t^2\right)\right\}, \tag{6}$$

where $t$ is the range time. $f_c$ is the carrier frequency. $T_p$ is the transmission time width. $B$ is the frequency modulation bandwidth. The sweep rate is defined as $K_r = B/T_p$.

　　After reflection, the signal that reaches the receiving antenna can be expressed as:

$$s_{R_m}(t - \tau_k) = \mathrm{rect}\left(\frac{t - \tau_k}{T_P}\right)\sigma(x,y)s_{T_n}(t - \tau_k), \tag{7}$$

where $\tau_k = \frac{r_{T_n}^A + r_{R_m}^A}{c}$, $c$ is the speed of light, $\sigma(x,y)$ is the scattering coefficient of targets, $k$ is the combination of receiving and transmitting channels. We use the dechirp technique to reduce the sampling rate by mixing the receiving and transmitting signals. The dechirp signal is given as follows:

$$s_{IF}(k,t) = s_{R_m}(t - \tau_k) * s_{T_n}^*(t) = \sigma(x,y)\exp(-j2\pi K_r t\tau_k)\exp(-j2\pi f_c\tau_k)\exp\left(j\pi K_r\tau_k^2\right), \tag{8}$$

where $*$ denotes complex conjugate. The first exponential term in Equation (8) is the distance dimension, which stands for the fast time information. The second exponential

term is the Doppler phase-frequency arising from the relative movement between the radar and the target. The last exponential term is the unique residual video phase (RVP) after dechirp, which is generally ignored because its impact on imaging quality is small.

After range compression, the signal model can be simplified as:

$$s(k) = \sigma(x,y)\exp\left\{ -j\frac{2\pi f_c}{c}\left( r_{T_n}^A + r_{R_m}^A \right) \right\}. \tag{9}$$

## 4. Array Arrangement Calculation and Design

### 4.1. EPC Spacing Calculation

The spacing of the EPCs affects the aperture length and azimuth resolution when the number of spatial sampling points is determined. As an essential parameter of the antenna, the beamwidth is utilized to calculate the EPCs spacing for the radar transceiver array.

The incident angle of the electromagnetic wave is defined as the angle between the incoming direction and the radar normal direction. According to the spatial Nyquist theory, the spatial separation of omnidirectional antennas is required to be less than a quarter wavelength. However, this constraint is looser for directional antennas because directional antennas have a certain beamwidth. The amplitude of the echo signal with a large incident angle is attenuated outside the main lobe. Therefore, we calculate the reasonable EPCs spacing according to the beamwidth of the directional antenna and the incident angle of the electromagnetic wave.

As shown in Figure 3, assuming that the incident angle of the electromagnetic wave is $\theta$ and the distance between the EPCs is $S$, the path difference from the signals to the two adjacent EPCs is $\triangle R = S\sin\theta$. The EPC spacing satisfies:

$$S \leq \left| \frac{\lambda}{4\sin\theta} \right|_{maxim}. \tag{10}$$

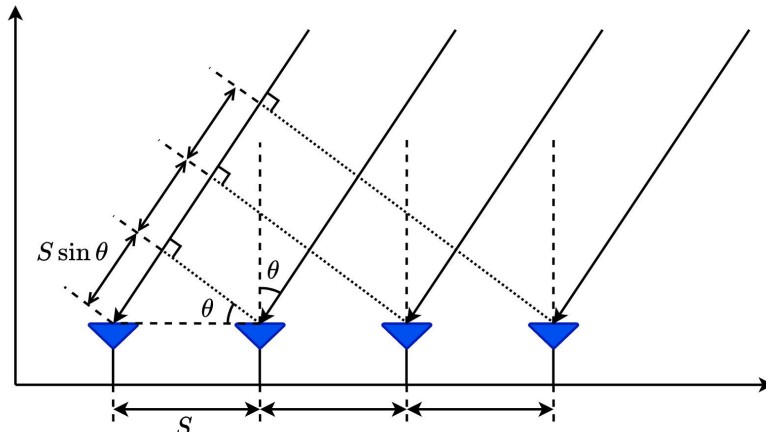

**Figure 3.** Spatial geometry of EPCs and electromagnetic wave incidence illustrations.

For omnidirectional antennas, $S$ is smaller than $\lambda/4$. For directional antennas, the appropriate EPC spacing S can be calculated according to the beamwidth. To reduce the ambiguity, the path difference between the EM wave and the neighboring EPCs needs to be $\lambda/4$.

### 4.2. Arrangement Design of Receiving and Transmitting Array

In the MIMO arrangement shown in Figure 4, the array is composed of three parts, including one row of transmitting arrays and two rows of receiving arrays. Among them, the transmitting and receiving array elements are arranged in parallel. $N \times M$ EPCs are generated by $N$ transmitting elements and $M$ receiving elements. The $M$ receiving array elements are divided into two groups, and the number of each group is $M/2$. The spacing

between the receiving array elements is $d$. The spacing between the transmitting array elements is $(M/2)d$. The distance between the generated EPCs is $d/2$. However, the length of the transmit array is $(N-1)(M/2)d$, and the length of the receiving array is:

$$\frac{(N+1)M}{2}d - d, \tag{11}$$

which is longer than the transmitting array length. The total length of the array is determined by the length of the receiving array.

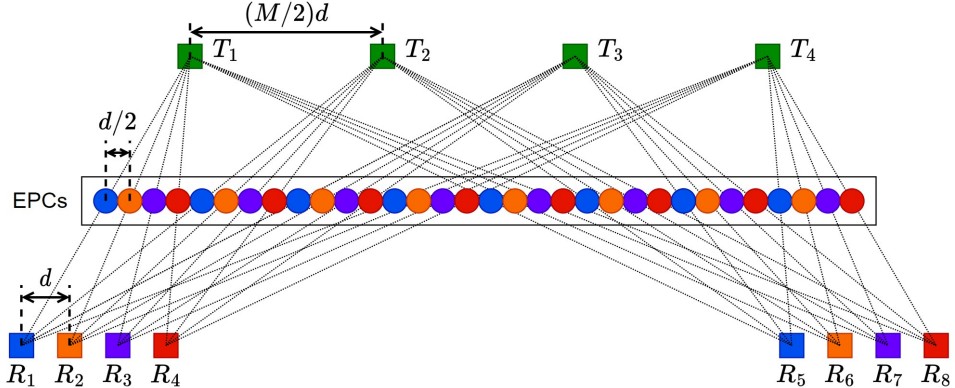

**Figure 4.** An illustration of a traditional array arrangement.

This arrangement can synthesize uniformly distributed EPCs. As a consequence of the small distance between the array elements, the variety of antenna choices is reduced, and the antenna design becomes more difficult. To solve this problem, we design a new arrangement, as shown in Figure 5. The arrangement of the receiving array elements is divided into four groups, where the spacing within the group is $2d$, and the spacing between groups is $3d$.

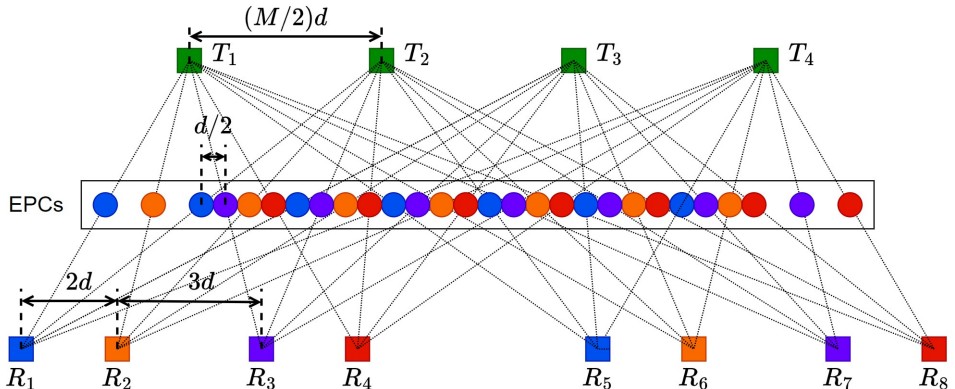

**Figure 5.** The illustration of innovative array arrangement.

According to the EPC principle, 16 transmitting elements and 16 receiving elements can form 256 EPCs. According to our proposed method, the 16 receiving elements are divided into four groups consisting of $R_1 - R_4$, $R_5 - R_8$, $R_9 - R_{12}$, and $R_{13} - R_{16}$, respectively. The spacing within each group is $2d$. The spacing between the first and second groups is $3d$, and the spacing between the third and fourth groups is $3d$ as well. The spacing between the second and third groups depends on the number of transmitting elements. With the proposed approach, 248 linear, equally spaced EPC arrays can be designed.

The new arrangement provides double the minimum spacing for the receiving elements, with only 3.2% aperture length loss, as shown in Table 1. Loss of aperture length affects azimuth resolution; however, this effect is negligible. Assume that the radar center

frequency is 30 GHz, and the spacing of receiving array elements is 5 mm according to the original method. The antenna spacing is so narrow that it is difficult to design the antenna. Considering our proposed method, the spacing can be improved to 10 mm. It is much easier to design a 10 mm width antenna than a 5 mm width antenna, especially the Horn antenna. On the other hand, the increased element spacing can be helpful for the design of the transmit or receive channel.

**Table 1.** Comparison of two arrangement parameters.

| Parameters | Traditional Arrangement | New Arrangement |
|---|---|---|
| Transmitting Elements Number $N$ | 16 | 16 |
| Receiving Elements Number $M$ | 16 | 16 |
| EPCs Number | 256 | 248 |
| EPCs Spacing $d$ | 2.5 mm | 2.5 mm |
| Minimum Elements Spacing | 5 mm | 10 mm |
| Aperture Length | 640 mm | 620 mm |

## 5. GB-SAR System Error Estimation

### 5.1. Signal Error Model

There are some errors in the practical application due to the special structure of MIMO arrays and a large number of array elements. The three types of error are shown in Figure 6. The first type of error is caused by the difference between the Tx/Rx channels, including the difference in the transmission path of the signal caused by circuit traces, switches, and amplifiers. The second type of error is the difference in amplitude between the T/R channels, including the amplification characteristics of the amplifier elements, transmission cable attenuation, splice losses, and antenna standing wave differences. The third type of error is the array elements position error during production and installation.

Transceiver array element position errors can be decomposed into three different directions: horizontal, vertical, and radial. The described errors will lead to amplitude and phase inconsistencies between channels, which will affect the focus of the image.

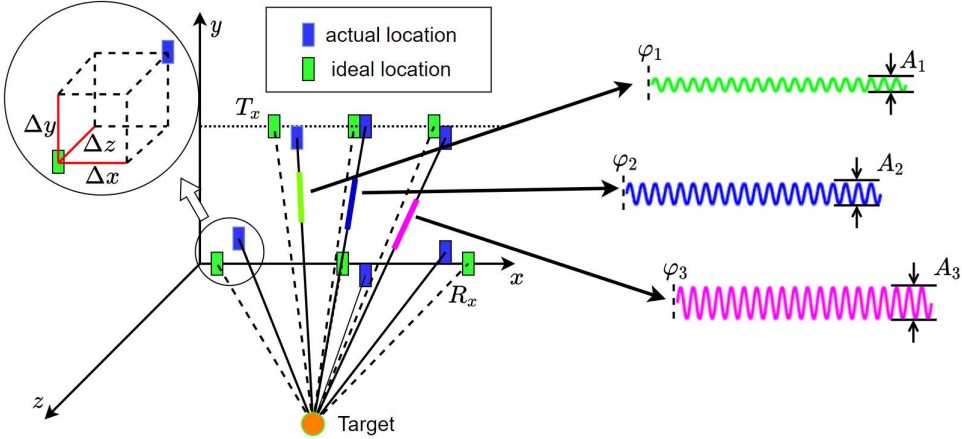

**Figure 6.** The illustration of three types of errors: magnitude error, initial phase error, and position error.

With the presence of array errors, the equation after range-directed compression is redescribed as:

$$s(n, m) = A_{T_n} A_{R_m} \sigma(x, y) \exp\left\{ j\left[ -\frac{2\pi f_c}{c}\left( \tilde{r}^A_{T_n} + \tilde{r}^A_{R_m} \right) + \phi_{0,T_n} + \phi_{0,R_m} + \phi_N \right] \right\}, \quad (12)$$

where

$$\widetilde{r}_{T_n}^A = \sqrt{(x_{T_n} + \Delta x_{T_n} - x_T)^2 + (y_{T_n} + \Delta y_{T_n} - y_T)^2 + (z_{T_n} + \Delta z_{T_n} - z_T)^2}, \quad (13)$$

$$\widetilde{r}_{R_m}^A = \sqrt{(x_{R_m} + \Delta x_{R_m} - x_T)^2 + (y_{R_m} + \Delta y_{R_m} - y_T)^2 + (z_{R_m} + \Delta z_{R_m} - z_T)^2}. \quad (14)$$

$\widetilde{r}_{T_n}^A$ is the distance between the actual position of the transmitting array elements and the target. $\widetilde{r}_{R_m}^A$ is the distance between the actual position of the receiving array elements and the target. $\Delta x_{T_n}$, $\Delta y_{T_n}$ and $\Delta z_{T_n}$ are the position errors of the $n$th transmitting array element in the three axis directions, respectively. $\Delta x_{R_m}$, $\Delta y_{R_m}$ and $\Delta z_{R_m}$ are the position errors of the $m$th receiving array element in the three axis directions, respectively. $A_{T_n}$ and $A_{R_m}$ represent the gains of the receiving chain and transmitting chain, respectively. $\phi_{0,T_n}$, $\phi_{0,R_m}$ represent the phase difference between the transmitting array elements and the receiving array elements due to the delay of the hardware circuit. $\phi_N$ is the random noise.

The position error along the three axes are $\Delta x$, $\Delta y$, and $\Delta z$. The error component $\Delta x$ in the azimuth direction affects the azimuth focus, and the component $\Delta y$ in the pitch direction affects the range focus. Since the antenna pitch and azimuth beams are narrow, the range component $\Delta z$ has a negligible effect on the pitch and azimuth focus. The range-direction position error will not cause the fluctuation of the distance unit. Therefore, only its influence on the initial phase is considered. Extending the position errors to three directions as follows:

$$s(n, m) = A_{T_n} A_{R_m} \sigma(x, y) \exp\{j[-\frac{2\pi f_c}{c}\left(r_{T_n}^A + r_{R_m}^A\right) - \frac{2\pi f_c}{c}(\Delta x_{T_n} + \Delta x_{R_m})\sin\theta$$
$$-\frac{2\pi f_c}{c}(\Delta y_{T_n} + \Delta y_{R_m})\sin\alpha - \frac{2\pi f_c}{c}(\Delta z_{T_n} + \Delta z_{R_m}) + \phi_{0,T_n} + \phi_{0,R_m} + \phi_N]\}, \quad (15)$$

where $\theta = \arctan\left(\frac{x_T}{r_T}\right)$ represents the incident angle of the target along the azimuth direction. $\alpha = \arctan\left(\frac{y_T}{r_T}\right)$ represents the incident angle in the pitch direction. $r_T$ represents the distance from the target to the radar.

As the special combination of channels, the channel combination of $T_n$ and $R_m$ is denoted by $k$ as follows:

$$s(k) = A_k \sigma(x, y) \exp\left\{j\left[-\frac{2\pi f_c}{c}r_{P_k}^A - \frac{2\pi f_c}{c}\Delta x_k \sin\theta - \frac{2\pi f_c}{c}\Delta y_k \sin\alpha - \frac{2\pi f_c}{c}\Delta z_k + \phi_{0,k} + \phi_N\right]\right\} \quad (16)$$

$r_{P_k}^A$, $\Delta x_k$, $\Delta y_k$ and $\Delta z_k$ indicate the distance after the channel combination and the position error in three directions. $\Delta z_k$ represents the component of the position error along the range in which the effects by the pitch and azimuth angles can be ignored. Combine this component with the initial phase $\phi_{0,k}$ as $\varphi_k$, we obtain:

$$s(k) = A_k \sigma(x, y) \exp\left\{j\left[-\frac{2\pi f_c}{c}r_{P_k}^A - \frac{2\pi f_c}{c}\Delta x_k \sin\theta - \frac{2\pi f_c}{c}\Delta y_k \sin\alpha + \varphi_k + \phi_N\right]\right\}. \quad (17)$$

During the processing, $\varphi_k$ is estimated as the overall initial phase.

*5.2. Channel Amplitude and Phase Correction*

In channel amplitude and phase estimation, the values of $A_k$ and $\varphi_k$ are obtained. Calculating the absolute value of Equation (17), we can obtain the expression:

$$A_k = \frac{|s(k)|}{\sigma(x, y)}. \quad (18)$$

The initial phase can be obtained as:

$$varphi_k = \phi_k + \frac{2\pi f_c}{c}r_{Pk}^A + \frac{2\pi f_c}{c}\Delta x_k \sin\theta + \frac{2\pi f_c}{c}\Delta y_k \sin\alpha - \phi_N, \quad (19)$$

where $\phi_k$ and $\frac{2\pi f_c}{c} r_{Pk}^A$ are known variables. $\Delta x_k$ and $\Delta y_k$ are unknown variables, which can be eliminated when $\theta$ and $\alpha$ are zero. $\phi_N$ is the random phase of each channel, the mathematical expectation is 0, and its effect on the initial phase is expected by calculating the mathematical expectation, i.e.,

$$\varphi_k = \mathbb{E}\left[ \phi_k + \frac{2\pi f_c}{c} r_{Pk}^A - \phi_N \right]. \tag{20}$$

*5.3. Array Elements Position Error Estimation*

When estimating the position errors of the array elements, a signal model is established according to the EPC principle, and multiple strong point targets are used for estimation. Under far-field conditions, it is assumed that the incident angle of the *i*th target is $\theta_i$. Then the phase of the echo obtained by the target at point *i* is:

$$\phi_{i,k} = -\frac{2\pi f_c}{c} r_{E_k}^{T_i} - \frac{2\pi f_c}{c}[(k-1)d + \Delta x_k] \sin\theta_i - \frac{2\pi f_c}{c} \Delta y_k \sin\alpha_i + \varphi_k. \tag{21}$$

$r_{E_k}^{T_i}$ is the two-way distance from the *k*th EPC to the target, then the phase difference between the two target echoes can be expressed as:

$$\Delta\phi = \phi_{1,k} - \phi_{2,k} = -\frac{2\pi f_c}{c}\left( r_{E_k}^{T_1} - r_{E_k}^{T_2} \right) - \frac{2\pi f_c}{c}[(k-1)d + \Delta x_k](\sin\theta_1 - \sin\theta_2) \\ - \frac{2\pi f_c}{c}\Delta y_k(\sin\alpha_1 - \sin\alpha_2). \tag{22}$$

Finding the expectation for both sides of the equation, we can obtain the following expression:

$$\mathbb{E}[\Delta\phi] = -\frac{2\pi f_c}{c}\mathbb{E}\left\{ \left( r_{E_k}^{T_1} - r_{E_k}^{T_2} \right) + [(k-1)d + \Delta x_k](\sin\theta_1 - \sin\theta_2) + \Delta y_k(\sin\alpha_1 - \sin\alpha_2) \right\}. \tag{23}$$

The mathematical expectation of the position error is close to zero. Therefore, when $\alpha_1 = \alpha_2$, $\sin\theta_1 - \sin\theta_2$ can be given by:

$$\sin\theta_1 - \sin\theta_2 = \frac{\mathbb{E}[\Delta\phi] + \frac{2\pi f_c}{c}\mathbb{E}\left[ r_{E_k}^{T_1} - r_{E_k}^{T_2} \right]}{-\frac{2\pi f_c}{c}\frac{k-1}{2}d}. \tag{24}$$

The value $\Delta x_k$ can be obtained by entering Formula (23). When $\theta_1 = \theta_2$, $\sin\alpha_1 - \sin\alpha_2$ can be obtained as follows:

$$\sin\alpha_1 - \sin\alpha_2 = -\frac{\mathbb{E}[\Delta\phi] + \frac{2\pi f_c}{c}\mathbb{E}\left[ r_{E_k}^{T_1} - r_{E_k}^{T_2} \right]}{-\frac{2\pi f_c}{c}}. \tag{25}$$

The value $\Delta y_k$ can be obtained.

## 6. Simulation and TD-MIMO GB-SAR Imaging Experiment

*6.1. Simulation*

Simulations are performed to verify the correctness of the estimation algorithm. The simulation parameters are shown in Table 2. A MIMO system is used for the simulation, which consists of 16 transmitting arrays and 16 receiving arrays. The system operates in FMCW mode with a dechirp signal. Priori errors are added to the ideal signal. The amplitude errors, phase errors, and position errors satisfy the uniform distribution of $U(0.5, 2)$, $U(-\pi, \pi)$ and $U(-1 \text{ mm}, 1 \text{ mm})$, respectively.

**Table 2.** Simulation parameters.

| Item | Value |
| --- | --- |
| Transmitting Elements Number $N$ | 16 |
| Receiving Elements Number $M$ | 16 |
| Wavelength $\lambda$ | 10 mm |
| Sweep Time | 20 μs |
| EPC Spacing $d$ | 2.5 mm |
| Sweep Bandwidth | 500 MHz |
| Sampling Frequency | 600 MHz |
| Detection Distance | 1 Km |

The simulation results are represented in Figure 7. There is no obvious main lobe without initial phase compensation. The target is unfocused, as shown in Figure 7a. With the presence of channel amplitude error, the side lobe is connected, and the level increases, as shown in Figure 7b. The main lobe has a slight deformation, while the side lobe has no obvious profile. The effect of position error on focus is shown in Figure 7c. Figure 7d shows the focus result after the error calibration. The simulation results show that the impact of the initial phase error on the focus is huge, which determines whether it can be focused, and the impact of amplitude error and position error on the focus is not negligible.

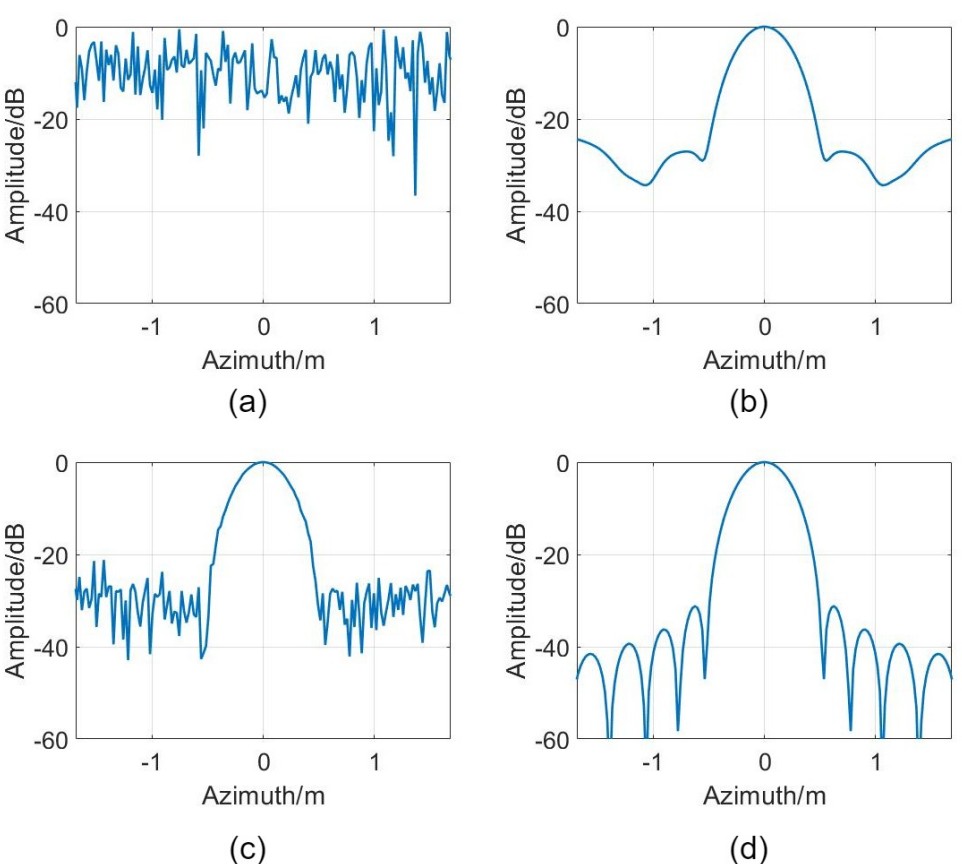

**Figure 7.** Simulation of the effect of different errors on focusing. (**a**) With initial phase error. (**b**) With amplitude error. (**c**) With position error. (**d**) After compensation.

*6.2. Experimental Results*

Practical experiments were carried out to verify the system performance and estimation method. For error calibration, a higher signal-to-noise ratio (SNR) target is necessary. To reduce the influence of other targets, the experiment scene is required to be as open as

possible. A piece of farmland in Xi'an city is used in this experiment. Figure 8a shows a photograph of the experimental scene. The scene is a depression area with a low-level center. Except for the green straw crops, all other areas in the scene are flat land.

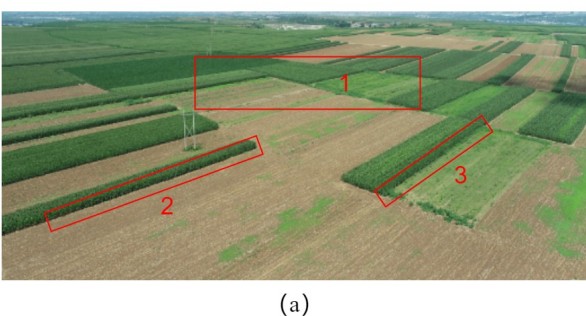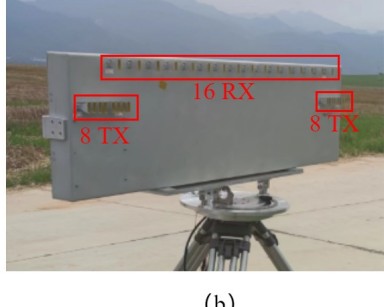

(a)　　　　　　　　　　　　　(b)

**Figure 8.** Experiment scene and system. (**a**) Low-lying and wide open fields. (**b**) GB-SAR system with 16 TX and 16 RX.

The GB-SAR system in the experiment is shown in Figure 8b. The system is equipped with 16 transmitting channels and 16 receiving channels. Transmitting elements are divided into 2 parts, each part consisting of 8 elements. The arrangement in each part is as described in Section 4.2. Transmitting channels and receiving channels are selected by a switching matrix. The time for a full channel scan is 248 times the ramp time. The system parameters are shown in Table 3.

**Table 3.** GB-SAR system parameters.

| Items | Value |
|---|---|
| TX Elements $N$ | 16 |
| RX Elements $M$ | 16 |
| Center Frequency | 30 GHz |
| Frequency Band | 1000 MHz |
| ADC sampling rate | 400 MSPS |
| Single Ramp Time $T$ | $\geq$20 μs |
| Time For A Single Full Scan | $\geq$4.96 ms |
| Detection Distance | 20~2000 m |

The imaging graph of the experiment scene is shown in Figure 9. As with the simulation results, the imaging is unfocused without phase calibration. In Figure 9a, the brighter lines are two calibrators placed at different distances. Regions 1 to 3 in the graph are blurred. The imaging graph after phase calibration is shown in Figure 9b. Since the radar is placed on the ground, the boundaries of crops in regions 1 to 3 are more easily observed than the entire crop area. After the initial phase calibration, the imaging effect is significantly improved.

The image after amplitude compensation is shown in Figure 9c, and the image background is clearer. The image after position error compensation is shown in Figure 9d, and the crop boundary is clearer than in Figure 9c. For further comparison, we take region 1 of the image for image entropy evaluation.

Image entropy was used to evaluate the imaging quality for each calibration, and the evaluation results are shown in Table 4. The results show that the image entropy decreased from 5.79 to 5.11. The target in the image is highlighted, and the noise in the background is reduced.

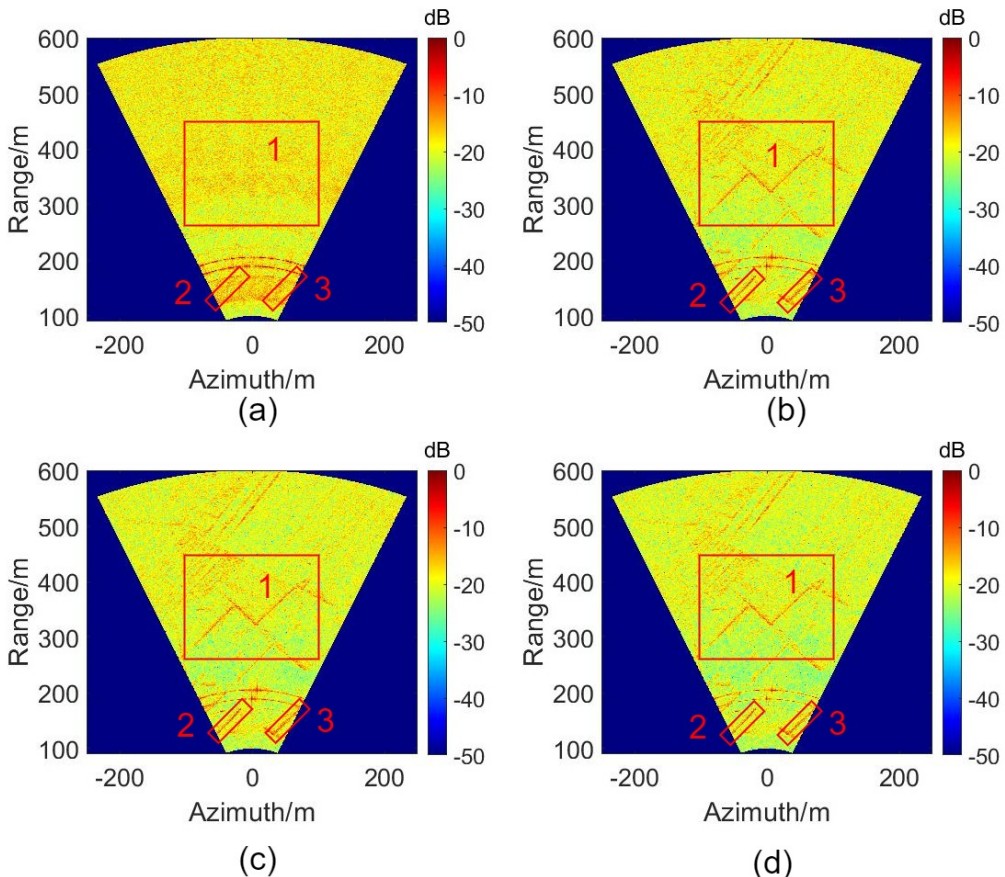

**Figure 9.** Experimental results. (**a**) BP imaging with initial phase error. (**b**) BP imaging with amplitude error. (**c**) BP imaging with position error. (**d**) BP imaging after compensation.

**Table 4.** Experimental results.

| Items | Without Calibration | Phase Error Calibration | Amplitude Error Calibration | Position Error Calibration |
|---|---|---|---|---|
| Image Entropy | 5.79 | 5.36 | 5.24 | 5.11 |

## 7. Deformation Detection Capability and Deformation Detection Experiments

### 7.1. Deformation Detection Capability Verification

In this subsection, two experiments are designed to demonstrate the high efficiency of acquiring data and the high-precision processing results. In the experiment, a corner reflector fixed on a programmable moving slider moves regularly shown in Figure 10. The programmable moving slider consists of a stepper motor and a precision screw. When stepping 42 times, the stepping motor will drive the screw to rotate 360°, which makes the corner reflector shift 1 mm forward. According to the mechanical structure above, the precision of the programmable moving corner reflector (PMCR) is better than 0.025 mm.

The first experiment verifies the high data acquisition rate of our ground-based radar system. The PMCR is programmed to make a periodic reciprocating motion, which firstly moves 1mm in the line of sight (LOS) direction and then moves 1mm in the opposite direction to complete the loop. In this way, the period of the round-trip movement is 100 ms without pause in the middle, and the maximum movement range is 1 mm.

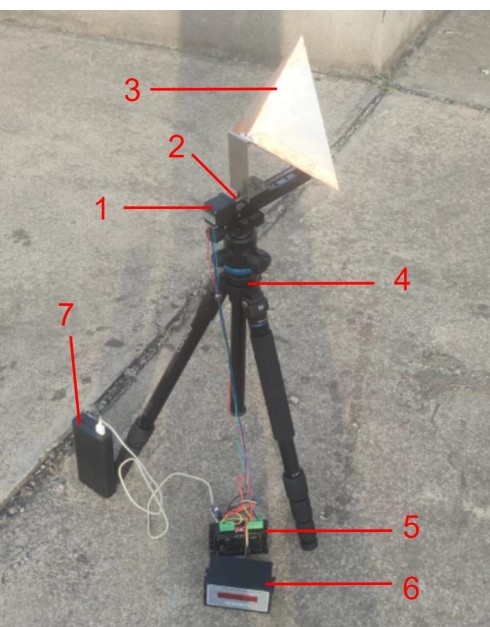

**Figure 10.** The illustration of the programmable moving corner reflector. Part 1 shows the stepping motor. Part 2 is a high-accuracy screw rod. Part 3 demonstrates the corner reflector. Part 4 express tripod. Part 5 is the controller. Part 6 shows the programmer. Part 7 is a battery.

The second experiment evaluates the accuracy of our system. Unlike the first continuous motion experiment, the PMCR is set to move intermittently in the second experiment. The reflector moves 0.2 mm along the LOS direction each time. After the movement, it will keep still for a certain time to complete the loop. After 5 loops, the PMCR returns to its starting position.

In our experiments, the size of the reflector is 14 cm, which is fixed on a tripod. The distance from our radar system to the PMCR is 120 m. Since the reflector motion direction is consistent with the LOS direction, the motion component in other directions is almost zero. It should be noted that the radar can observe the shift of each point in the scene, so our reflector can be served as an example for our observation experiment to test the high accuracy.

The data processing procedure includes system calibration and radar imaging. Interest points in the image are selected for observation. The measured shift is obtained by analyzing the phase change of an interesting point with time. The curve in Figure 11a approximates a triangular wave, but it is different from the standard one at the corner. The arc-shaped deformation differences at the corners are caused by the motion gaps of the stepper motor and the high-precision screw rod. The acquisition time of each image is 10 ms. According to the Nyquist sampling theorem, the radar can detect vibrations with a max frequency of 50 Hz. To achieve the same data acquisition rate, the conventional ground-based SAR system on the sliding rail should move several hundred meters per second.

Figure 11b shows the accuracy of the ground-based radar in detecting displacement. The corner reflector vibrates slightly during the pause time, which is caused each time there is a sudden stop. In the presence of the vibration, our radar measurement accuracy can be better than 0.04 mm.

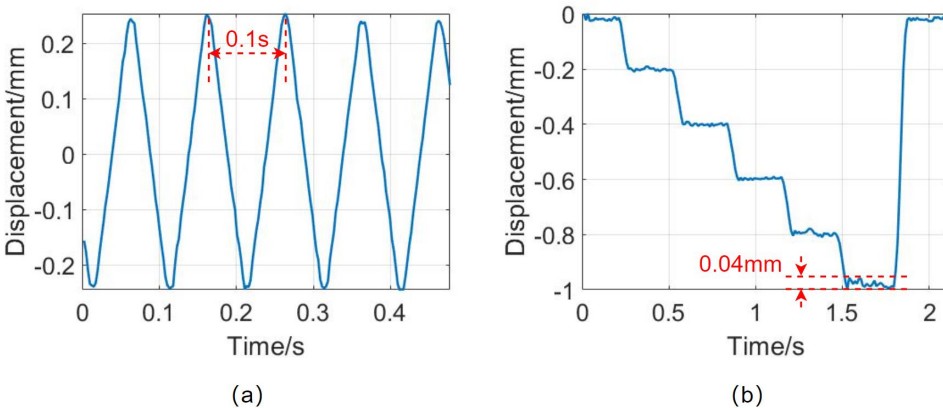

**Figure 11.** (**a**) The measured shift of the reciprocating motion. (**b**) The measured shift of the step motion.

### 7.2. Bridge Deformation Detection Experiment

Figure 12 illustrates the geometry of bridge deformation measured by our radar. The bridge deformation direction is vertical to the bridge floor. The pitch angle between the radar LOS direction and horizon direction is $\beta$. However, the displacement measured by our radar system is the bridge deformation projected on the radar LOS direction. Equation (26) shows the deformation of the bridge in the vertical direction.

$$d_p = \frac{d_{LOS}}{\sin \beta} \tag{26}$$

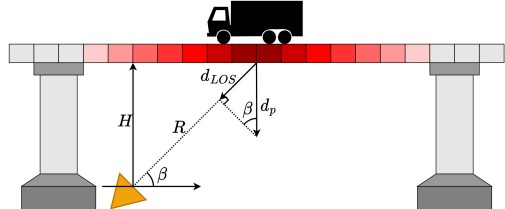

**Figure 12.** The illustration of bridge deformation detection model.

To further verify the generality of our algorithm, a practical scenario experiment was carried out on the Hongqi Canal Bridge in Xi'an. The GB-SAR system detects the deformation of the bridge to replace the original sensors. Furthermore, the system can detect the deformation at every point of the bridge simultaneously. Four points are selected for analysis in Figure 13a named points 1 to 4. The corresponding radar image result of these points is shown in Figure 13b. In the experiment, the GB-SAR is placed on the riverbank shown in Figure 13a. Large cargo trucks pass on the bridge from time to time. After 20 min of measurement, about 1900 images are collected. The bridge deformation information of points 1–4 is in Figure 14 when the large truck passes.

Figure 14 shows the deformation results of all points at different times. At 11:16:03, the truck passes point 1, and the deformation is about 2 mm. As shown in Figure 14a, the red circle represents the deformation, while the other blue areas represent the remaining points, which are stable at this time. At 11:16:05, the truck passes Point 2, the deformation is about 2 mm. The red area in Figure 14a becomes blue, which means the deformation disappears, while the red circle moves from point 1 to point 2 in Figure 14b because the truck passes here at this time. The same phenomenon occurs when the truck passes points 3 and points 4 corresponding to 11:16:07 and 11:16:09.

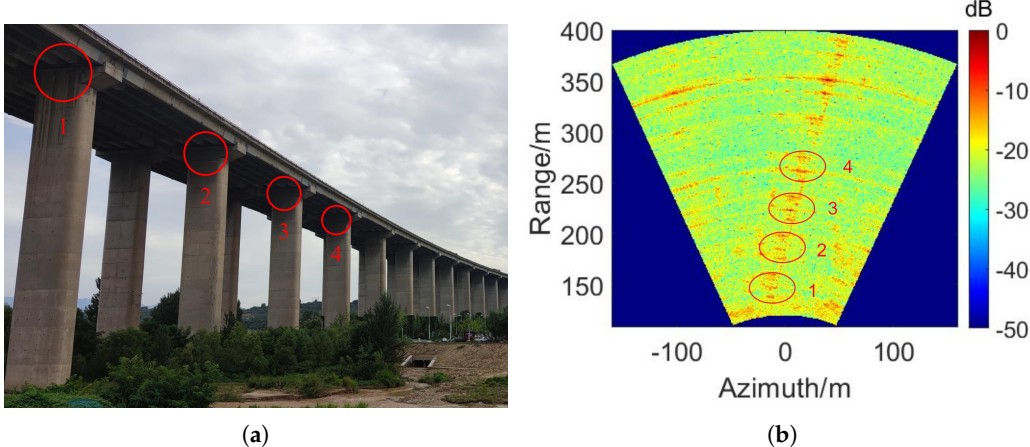

**Figure 13.** High-way bridge experiment scenario. (**a**) High-way bridge. (**b**) Imaging results.

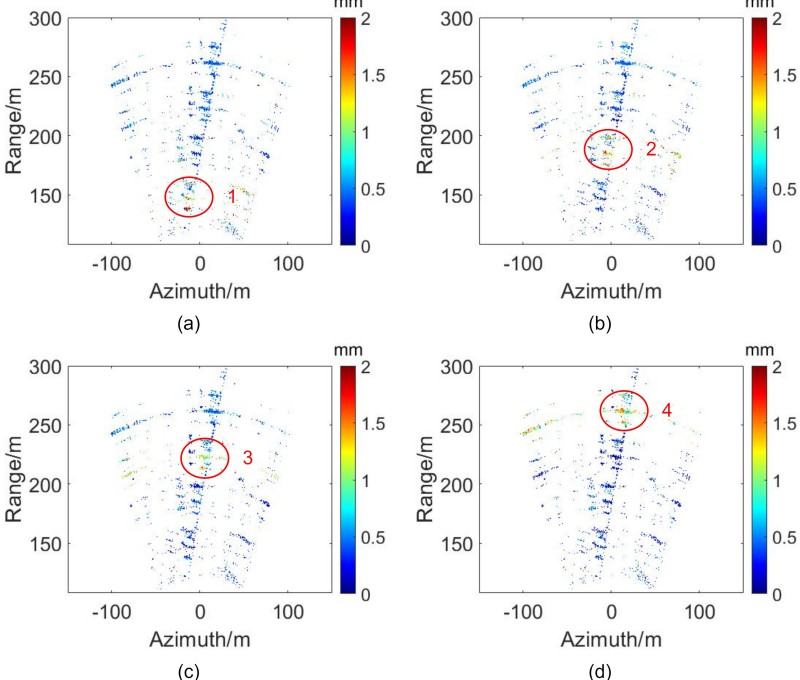

**Figure 14.** The bridge deformation diagram at different times on 27 August 2021. (**a**) The deformation at 11:16:03. (**b**) The deformation at 11:16:05. (**c**) The deformation at 11:16:07. (**d**) The deformation at 11:16:09.

Figure 15a–d corresponds to the red circles under the bridge in Figure 13, respectively. Figure 15 illustrates that the horizontal ordinates of the valley value are 8, 10, 12 and 14 s, respectively, which indicates the truck passes these points successively. The maximum downward deformation of the bridge is about 2 mm. When the truck leaves the point, the bridge recovers from the downward deformation. The deformation and recovery processes of the bridge are completed quickly in 2 s. Around the valley value, there appears a slighter bridge vibration of 1mm when the truck passes and leaves the point.

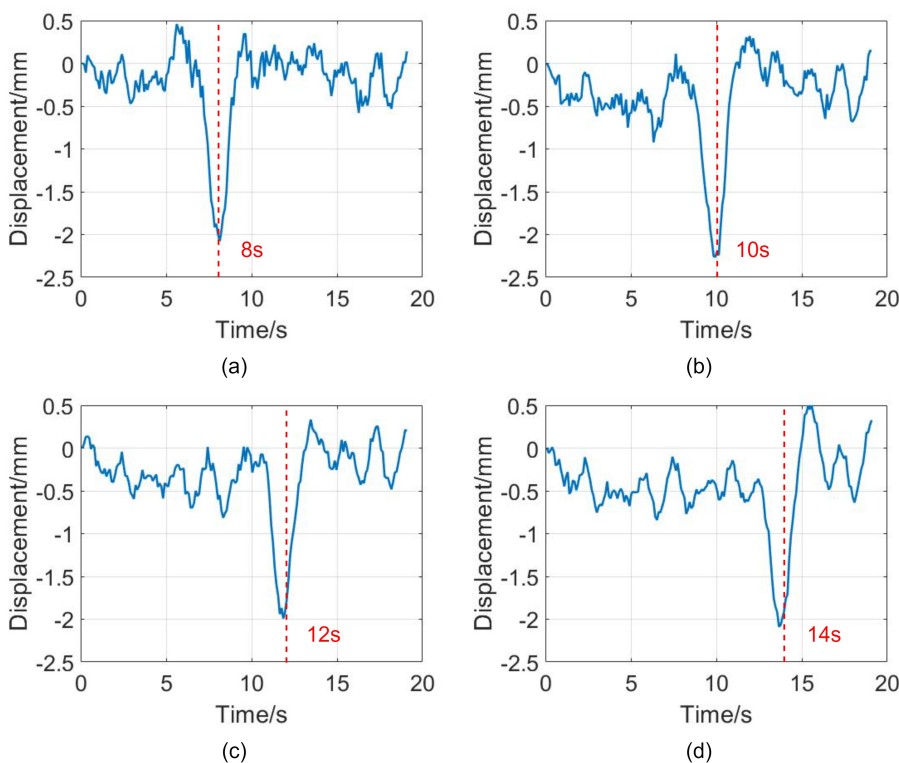

**Figure 15.** Deformation of different positions is measured at 11:16 on 27 August 2021. (**a**) represents the displacement of point 1. (**b**) represents the displacement of point 2. (**c**) represents the displacement of point 3. (**d**) represents the displacement of point 4.

Figure 16 shows the deformation curve and frequency spectra of the bridge when a large number of vehicles pass by. Figure 16a shows that the maximum deformation of point 1 is 3.3 mm. As illustrated in Figure 16b, the peak value of the corresponding spectrum is at 2.8 Hz except for zero frequency. Frequencies below 1Hz can be ignored because there is no apparent peak. Figure 16c shows the deformation curve for point 2 with a maximum value of 4.1 mm. The spectrum peak is the same as point 1, as shown in Figure 16d. Figure 16e shows the deformation curve of point 3, and the maximum deformation is 3 mm. The spectrum peak is at 2.8 Hz, as shown in Figure 16f. Figure 16g shows that the maximum deformation of point 4 is 2.9 mm. The spectrum peak is at 2.7 Hz, as shown in Figure 16h. The maximum deformation value is 2.9 to 4.1 mm from points 1 to point 4, while the frequency spectrum peak is around 2.8 Hz.

In addition to heavy vehicles, environmental factors such as temperature, humidity and atmosphere can affect the measurement of the bridge. Since the measurement time is short, the environmental change is ignored. The wind can make the bridge swing, especially for long bridges. The wind is slight during the experiment, and its effect on bridge deformation is not considered. Some conclusions can be drawn from the deformation measurement results. Firstly, when a truck passes, the bridge will deform significantly. Around the deformation valley value, there are some slight vibrations. Secondly, bridge deformation increases when multiple vehicles pass simultaneously. Thirdly, the bridge deformation may vary from region to region, but the vibration frequency may tend to be stable. Finally, the radar system can measure the deformation at each point on the bridge.

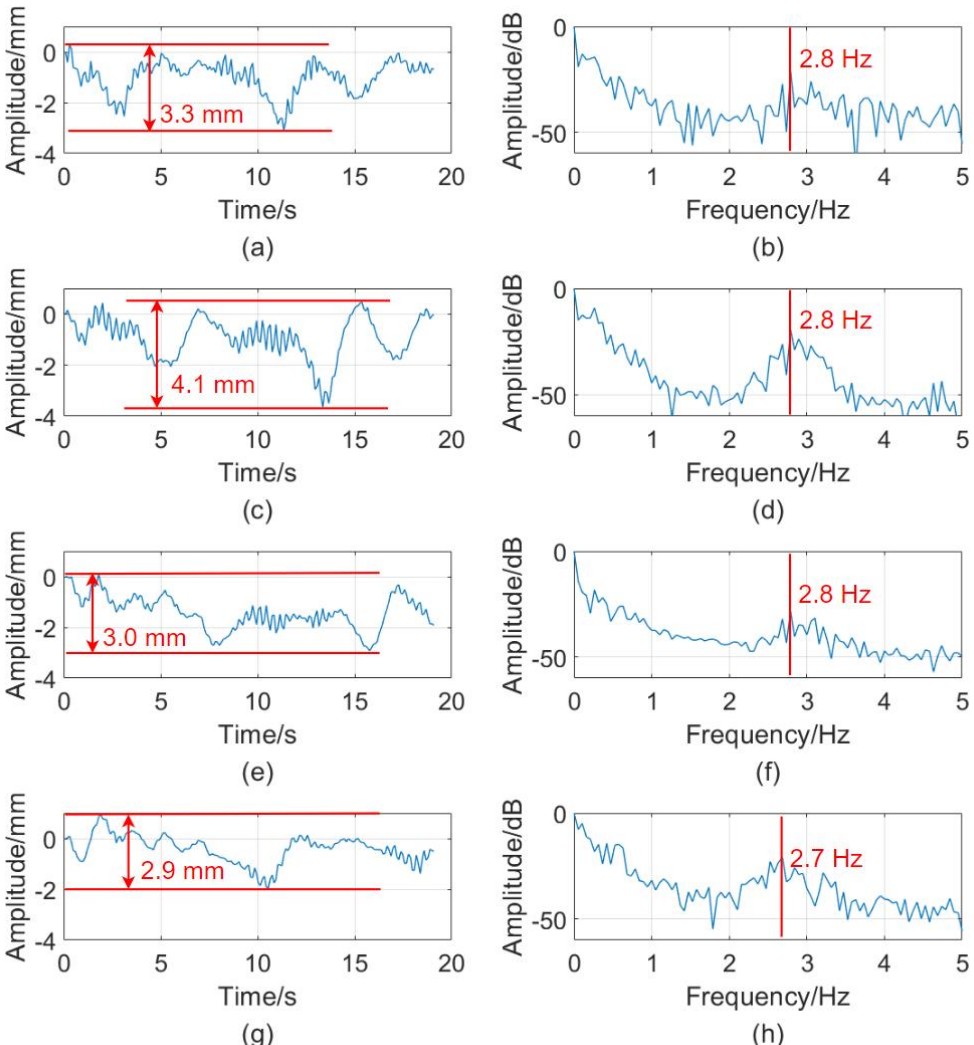

**Figure 16.** Measurement results of a large number of vehicles passing by at 11:10 on 27 August 2021. (**a**,**b**) show the deformation curve and spectrum of point 1. (**c**,**d**) show the deformation curve and spectrum of point 2. (**e**,**f**) show the deformation curve and spectrum of point 3. (**g**,**h**) show the deformation curve and spectrum of point 4.

## 8. Discussion

To increase the antenna placement spacing, a new MIMO transceiver array arrangement is designed. Compared with the previous arrangement, the minimum spacing of the receiving array elements is improved from 5 to 10 mm, and the aperture length loss is 3.2% for 16 transmitting and 16 receiving arrays. The experimental results in Section 6.2 show the effectiveness of this arrangement.

The amplitude errors, phase errors, and position errors during the production and installation are inevitable due to the special structure of MIMO arrays. The effect of these errors on imaging is analyzed by simulation. In the simulation in Figure 7, the target is unfocused without initial phase compensation. The channel amplitude error affects the level of the sidelobe, and the position error affects the noise floor of the imaging. The experiment results in Figure 9 show BP imaging is unfocused without calibration. After error compensation, the image entropy decreases from 5.79 to 5.11, which demonstrates the effectiveness of the method. In contrast to other methods, this method has a short estimation time without an iterative process. The phase calibration and amplitude calibration are adapted to any MIMO form of radar. The position error estimation method is adapted to

radar systems with small pitch angles. The disadvantage is the high requirement for the position of the scattering point and the scattering intensity.

In addition, a programmable moving reflector is used in our experiment to verify the system's capability in bridge vibration detection. In Figure 11, the measurement results prove the accuracy of this radar system up to 0.04 mm. Theoretical measurement rates reach up to 100 Hz. Experimental results of this radar system for bridge deformation and vibration measurements are presented in Figures 14 and 15. The vibration frequency and vibration amplitude of the bridge are analyzed in Figure 16. The measurement results show that the deformation varies with the gravity applied to the bridge.

## 9. Conclusions

We propose a new arrangement method that redesigns the position of the transceiver array elements to double the minimum element spacing. For this particular array structure, we propose an estimation method that decomposes the position error. This method divided position errors into three directions. The distance direction position error is regarded as a part of the initial phase. The position errors in the pitch and azimuth directions are estimated by strong reflection points. The experiment results demonstrate the validity of the arrangement and the feasibility of the error estimation method.

The GB-SAR system adopts TDM MIMO technology in its structure, which has data acquisition advantages compared with the mechanical GB-SAR system. The experiment proves that this GB-SAR has a high data acquisition rate, and the measurement accuracy can reach 0.04 mm. Bridge monitoring experiments have proven that the radar can monitor high-frequency vibrations of building structures such as bridges. As the data acquisition rate increases, the data processing rate also needs to be increased. In the future, we will focus on fast data processing algorithms to increase the image generation rate.

**Author Contributions:** Conceptualization, Z.Z. and Z.L.; methodology, Z.Z., Z.L. and Z.S.; software, Z.Z. and L.Q.; validation, Z.Z.; formal analysis, Z.Z. and Z.S.; investigation, Z.Z.; resources, Z.L. and Z.S.; data curation, Z.Z.; writing—original draft preparation, Z.Z. and F.T.; writing—review and editing, Z.L., F.T., Z.S. and H.T.; visualization, Z.Z. and L.Q.; supervision, Z.L., Z.S. and H.T.; project administration, Z.Z. and Z.L.; funding acquisition, Z.L. and Z.S. All authors have read and agreed to the published version of the manuscript.

**Funding:** This work was supported by the National Natural Science Foundation of China (Grant No. 62031005).

**Data Availability Statement:** Not applicable.

**Acknowledgments:** The authors gratefully acknowledge the helpful comments and suggestions of the reviewers of this manuscript.

**Conflicts of Interest:** The authors declare no conflict of interest.

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
