# Peer review of "A Novel GB-SAR System Based on TD-MIMO for High-Precision Bridge Vibration Monitoring"

_remotesensing, doi:10.3390/rs14246383_

Round 1
Reviewer 1 Report
To increase the place spacing of the antenna, the authors propose a novel arrangement method for the TD-MIMO array. In addition, a new error estimation method is proposed to improve imaging quality. The authors verify the effectiveness of the arrangement method and the error estimation algorithm through reasonable experiments. In the paper, the work is complete and the method is correct. My comments are listed as follows, which shall be addressed in the revised version of this paper.
1. In Table 1, the minimum element spacing is increased from 5mm to 10mm. The advantages of the new arrangement should be briefly described. The aperture length is reduced from 640mm to 620mm. What is the impact of a 3.2% loss of the aperture length?
2.In the error estimation and calibration, what are the requirements for the selection of the scene and reflection points?
3. What is the information represented by λ in Equation 5?
4. Some writing mistakes should be corrected, such as 'n' and 'm' in P4, L129 should be used subscripts, the title of Figure 3 should be described as “EPC ...”, etc.
Reviewer 2 Report
High-precision bridge vibration monitoring very well presented with mathematical modeling discussed in early, as specification mention in table 3 with 16 pairs of TX/RX, tally to figure 13 (a) which bridge experiment, does all bridge pillars occupied with sensors ? because in the figure just indicate several number. If no, is all number of pillars installed with sensor ?
While figure 13 (b) mention in discussion the deformation due to heavy vehicle passing, how many factors caused for bridge deformation beside heavy vehicle ?
Reviewer 3 Report
The present work regarding the use of a GB-SAR MIMO for bridge monitoring is well written and presented. The topic covered is currently one of the most interesting among the new GBSAR developments, as confirmed by recent publications such as:
Baumann-Ouyang, A.; Butt, J.A.; Salido-Monzú, D.; Wieser, A. MIMO-SAR Interferometric Measurements for Structural Monitoring: Accuracy and Limitations. Remote Sens. 2021, 13, 4290. https://doi.org/10.3390/rs13214290
which I suggest to mention in the introduction.
In my opinion, there are some fixes and improvements that should be made prior to publication:
Equation (1) and following: capital R is used both to indicate RX and distance. This is accettable but I would suggest to use different letters as for example, lowercase r for distance.
Equation (7): index k is introduced instead of (m,n) but it is defined several lines below (line 153). I would suggest to define it immediately after its introduction.
Line 191 antenna spacing configuration is showed for 4 TX and 8 RX, but it is not immediately clear how to extend it to 16 TX and 16 RX. I suggest briefly illustrating what is the RX arrangement for 16 elements.
Equation (12) and following: index c is missing from carrier frequency
Equation (15) and following: index n is used both for transmitting elements and noise. To avoid confusion, I would suggest to use another index for noise (e.g. capital N)
Line 229: In my opinion, the correct formula for \theta angle is \theta = asin(x/r), which for small angles is equivalent to the written one.
Line 230: same for \alpha angle: \alpha = asin(y/r)
Figure 7: it seems to me that no tapering window is applied in the azimuth focusing. In fact, after error compensation, the first sidelobe level is 13 dB.
Usually a tapering windows are used in GBSARs, as for example kaiser window with beta between 3 and 6 . To evaluate the goodness of the calibration it is also necessary to use a tapering window and check the level of the sidelobes. Also in figure 9 the two strongest targets shows strong sidelobes even after error compensation. I strongly suggest to test the calibration effectiveness with a tapering window.
Section 6.2 The GB-SAR instrument used for the measurements is not adequately presented. Apart from the Table 3, there is neither a brief description nor a photo of the system. Given the subject of the paper, I believe a more detailed presentation of the GB-SAR system is mandatory.
Line 296: 'corner reflector' is a more common term instead of 'angular reflector'
